# Cosmetic Treatment Using Botulinum Toxin in the Oral and Maxillofacial Area: A Narrative Review of Esthetic Techniques

**DOI:** 10.3390/toxins15020082

**Published:** 2023-01-17

**Authors:** Sung Ok Hong

**Affiliations:** Department of Oral and Maxillofacial Surgery, Kyung Hee University College of Dentistry, Kyung Hee University Hospital at Gangdong, Seoul 05278, Republic of Korea; catherine.so.hong.sleepdoc@gmail.com; Tel.: +82-02-440-7517

**Keywords:** botulinum toxin, Botox, cosmetic treatment, esthetic treatment, maxillofacial, dentist, injection, BoNT, non-invasive, face, wrinkles, aging, beauty, facial anatomy, toxin, rhytides

## Abstract

Botulinum toxin (BoNT) is an anaerobic rod-shaped-neurotoxin produced by Clostridium botulinum, that has both therapeutic and lethal applications. BoNT injection is the most popular cosmetic procedure worldwide with various applications. Patients with dynamic wrinkles in areas such as the glabella, forehead, peri-orbital lines, nasal rhytides, and perioral rhytides are indicated. Excessive contraction of muscles or hyperactivity of specific muscles such as bulky masseters, cobble stone chins, gummy smiles, asymmetric smiles, and depressed mouth corners can achieve esthetic results by targeting the precise muscles. Patients with hypertrophic submandibular glands and parotid glands can also benefit esthetically. There are several FDA-approved BoNTs (obabotuli-numtoxinA, abobotulinumtoxinA, incobotulinumtoxinA, letibotulinumtoxinA, prabotulinumtox-inA, daxibotulinumtoxinA, rimbotulinumtoxinB) and novel BoNTs on the market. This paper is a narrative review of the consensus statements of expert practitioners and various literature on the injection points and techniques, highlighting both the Asian and Caucasian population separately. This paper can serve as a practical illustrative guide and reference for optimal, safe injection areas and effective doses for application of BoNT in the face and oral and maxillofacial area. The history of BoNT indications, contraindications, and complications, and the merits of ultrasonography (US)-assisted injections are also discussed.

## 1. Introduction

Facial beauty, though slightly different in each ethnicity, tends to have a universal rule. A balanced, symmetrical, oval face with harmonious features is perceived as attractive and youthful [1,2,3]. Esthetic concerns are different by age groups with younger groups focusing more on a smooth contour and volume reduction, while older groups focus on wrinkles and improving volume loss [3]. Facial expression muscles are different from other muscular parts of the body because they have soft tissue attachments to the skin through the superficial musculoaponeurotic system, rather than to the bone [4]. Since contracting superficial skin moves in coordination with the underlying muscles, dynamic wrinkles perpendicular to the muscles develop [4,5]. These developed dynamic rhytides lead to atrophy of the dermis and pleating in the skin, which may give an impression of an angry, sad, or aging face. Botulinum neurotoxin (BoNT) can significantly soften dynamic wrinkles and even provide a lifting effect [6,7]. BoNT injection is the most popular cosmetic procedure worldwide with various applications. It can treat not only facial wrinkles but also a wide range of dermatologic, ophthalmologic, oral and maxillofacial, neurologic, urologic, and gynecologic conditions [8,9,10,11]. The reversible characteristics and versatile applications of BoNT make it widely preferable for both cosmetic and therapeutic treatment.

BoNT is a neurotoxin produced by the anaerobic rod-shaped bacterium *Clostridium botulinum*, and has both medical and lethal applications [12]. It is the most potent biological toxin known with an estimated lethal dose at 0.09 to 0.15 µg (IV or IM), 0.70 to 0.90 µg (inhalation), and 70 µg (PO) [11,13]. *C. botulinum* produces seven BoNT serotypes (A, B, C, D, E, F, G) [14,15]. An eighth serotype (X) was identified from *C. botulinum* strain 111 using bioinformatics techniques [15,16]. The most potent serotype is A, followed by B and F [17,18]. BoNT serotypes A and B are the only ones that have been manufactured for clinical use [7]. The effects of BoNT appear after approximately 2–3 days on rhytides and 3 weeks on larger muscles, and last between 3 and 6 months [19,20,21]. 

In 1829, a German medical officer and poet named Justinus Kerner (1786–1862) reported lethal food poisoning due to rotten sausages (*botulus*, Latin for sausage). In 1897, Emile von Ermengem investigated an outbreak of botulism due to raw ham in Belgium and found spore-forming anaerobic bacteria, *Bacillus botulinum* [19]. The outbreak of World War I and II weaponized BoNT by investigating the effect on muscle contraction [22]. In 1944, Edward Schantz cultured and isolated *C. botulinum*, and later in 1949, Arnold Burgen discovered the mechanism of how BoNT works by blocking acetylcholine release from the neuromuscular junction leading to muscular paralysis [11,23].

The first suggestion of using BoNT for therapeutic use was in 1973. An American ophthalmologist Alan B. Scott published the results of strabismus treatment using BoNT on monkeys. In 1977, he became the first person to use BoNT to treat blepharospasm and strabismus, and later founded the company Oculinum [24]. In 1987, Carruthers and Carruthers observed that BoNT treatment in patients with blepharospasm enhanced their glabellar lines and reported its cosmetic effect [25]. Soon after the United States Food and Drug Administration (FDA) approved BoNT in 1989, Oculinum was bought by Allergan in 1991, and Botox^®^ was manufactured. After this, the FDA has approved BoNT for various cosmetic and non-cosmetic applications. Cosmetic applications for glabellar lines in 2002, severe axillary hyperhidrosis in 2004, migraines in 2010, lateral canthal lines in 2013, forehead lines in 2017 and sialorrhea in 2018 [9,26].The purpose of this review is to present an overview of BoNT and demonstrate the optimal, safe injection areas and effective dose for application of BoNT in the oral and maxillofacial area. This practical guideline depicts the anatomy of the facial muscles relative to the BoNT injection points and includes a narrative literature review.

## 2. Materials and Methods

This review explores quantitative data using a systematic search on studies depicting dose range and injection points from expert consensus opinions. A literature search was conducted using MEDLINE and PubMed electronic databases for the period from January 2002 to December 2022. Manual study searches were also performed using the reference list of articles included during the search process.

The inclusion criteria were as follows:Studies in the English language;Include dosage of BoNT and/or number of injection points;Be performed in human adults with no diseases;Given BoNT injection of the face for cosmetic or esthetic purpose;The literature is limited to expert consensus.

Exclusion criteria were as follows:Studies using BoNT for other reasons than cosmetic/aesthetic: i.e., stroke, neurogenic, bladder management, spasticity, sialorrhea, bruxism, strabismus, myofascial pain, TMJ, or obesity;Studies using BoNT for cosmetic/aesthetic reasons in other parts of the body or other reasons: i.e., skin texture, scar, androgenic alopecia, neck wrinkles, trapezius hypertrophy, or calf hypertrophy;Studies that investigated only a single muscle.

The search included the following keywords used in different combinations: “botox”, “botulinum toxin”, “consensus”, “aesthetic”, “cosmetic”, “systematic review”, and “face”. The term “botox consensus” retrieved 202 studies, “botulinum toxin” AND “consensus” retrieved 315 studies. The search term “face botulinum toxin consensus” resulted in 56 studies, “esthetic botulinum toxin” AND “consensus” resulted in 54 articles, and “cosmetic botulinum toxin consensus” resulted in 42 studies. The term “botox systematic review” retrieved 421 articles.

## 3. Results

Titles and abstracts were screened, the eligibility criteria were applied, and full text analysis was performed in relevant publications. Thirty-three full text articles were screened for eligibility, of which six were excluded because of a lack of BoNT dosages and type of paper. A total of 27 articles were included in the literature review. The injection dosage and points proposed in this paper are based on the findings from such various reviews.

The BoNT dose and injection points from the views of expert opinions are summarized in Table 1, Table 2 and Table 3. A total of 27 consensus articles were reviewed after retrieving the data through a systematic search. Of the 27 studies, 4 were specified towards Asians, and therefore organized into a separate table (Table 3) [27,28,29,30]. Of the 23 other studies, those by Fagien et al. [31] and by Bertossi et al. [32] were each separately updated by the same group and therefore were excluded from the summary. Carruthers et al. [33] updated their consensus in 2008 and again in 2013 [34,35]. The initial 2004 consensus was grouped with the 2008 consensus, but the 2013 consensus was summarized because of slightly different parameters. Studies by Raspaldo et al. [36] and Gassia et al. [37] were identical to the 2010 consensus from Raspaldo et al. [38,39] and excluded from the summary tables. Most of the studies used onabotulinumtoxinA units, but three studies using different units of abobotulinumtoxinA were excluded from the review [40,41,42]. Three other studies using different BoNTA were included. Studies by Sundaram et al. [28] and Yutskovskay et al. [43] used incobotulinumtoxinA, while Ahn et al. [27] utilized Medytox. A total of 16 consensus papers were summarized in this narrative review (Table 1, Table 2 and Table 3).

### 3.1. Cosmetic Indications and Contraindications

Elucidating the cosmetic indications and contraindications of BoNT is crucial to maximize patient satisfaction and minimize complications. Patients with dynamic wrinkles, which are formed during facial expression are indicated [4]. Areas with dynamic motion such as the glabella, forehead, peri-orbital lines, nasal rhytides, and perioral rhytides are common injection areas [7,54,55]. Patients with static wrinkles that are present at rest can also be candidates. However, the results are less dramatic and additional treatment sessions or additional esthetic procedures, such as fillers, may be required to achieve ideal outcomes. Excessive contraction of muscles or hyperactivity of specific muscles can also cause non-cosmetic appearances. Patients with bulky masseters, cobble stone chins, gummy smiles, asymmetric smiles, depressed mouth corners, brow ptosis, flaring nostrils, depressed nasal tips, temporal hypertrophy, and trapezius hypertrophy can achieve esthetic results by targeting the precise muscles [7,22]. Patients with hypertrophic submandibular glands and parotid glands can also benefit esthetically in select cases [56].

Contraindications include immunocompromised patients with neuromuscular disorders, such as myasthenia gravis and Lambert–Eaton syndrome; neurodegenerative diseases such as amyotrophic lateral sclerosis; and pregnancy and breast-feeding [4,19]. BoNT is also contraindicated in patients with unrealistic expectations, body dysmorphic syndrome, keloidal scarring, active dermatoses or infection in the treatment area, motor weakness in the treatment area (Bell’s palsy), and allergic reactions to BoNT constituents (cow’s milk protein allergy using abobotulinumtoxinA) [57]. Theoretical drug interactions between BoNT and aminoglycoside antibiotics, quinidine, calcium channel blocker, magnesium sulfate, succihylcholine, and polymyxin exist [22,58,59].

### 3.2. Cosmetically Used Botulinum Toxin Products

Currently, there are several popular FDA-approved (six BoNTA, one BoNTB) and non-approved toxin products on the market (Table 4) [8,12,60,61]. BoNTs vary in composition, complex size, molecular and chemical properties, supply form, and immunogenicity. Pure BoNT is manufactured as 150 kDa proteins that bind in different quantities to neurotoxin-associated complexing proteins (NAPs) to form high-molecular-weight progenitor complexes [8,62]. Commercial BoNT products have different complements of NAPs, which lead to the formation of various molecular weights and three-dimensional structures [62]. Excipients are added to ensure long-term storage and minimize inactive neurotoxins because they might not interact with nerve cells but are recognized by the immune system [63].

### 3.3. Facial Target Muscles and Injection Doses

Accurately administered injections and detailed knowledge of the anatomy to determine the correct depth and injection points are important factors for minimizing complications. The relevant anatomy including the muscles involved and amount of injection, is shown in Table 5 [20,28,44,55,60]. Thorough comprehension of the facial musculature and adjacent anatomy is essential before treatment initiation. Diffusion of BoNT differs according to the amount of mixed saline. However, in general, 1–5 mL of saline is mixed with 100 U of BoNTA, which results in diffusion distances 2.5–3 cm in diameter [4,20,64]. Before injection, good lighting, marking injection points, prefilling syringes with the required dose, and pre-photography are beneficial.

### 3.4. Injection Techniques of BoNT in the Face

BoNT injection depth and amounts differ within each area of the face. Intramuscular injections act on the neuromuscular end plate, leading to denervation of the muscles and subsequent relaxation [19]. Injections can be subdermal in areas where the muscles are very superficial, such as the orbicularis oculi and frontalis muscles. The maximum injection dosage should be injected into the muscle origin, where action is the strongest. Muscles with fibers inserted into the skin can be injected intradermally. Moreover, superficial fibers can be targeted subdermally [4].

#### 3.4.1. Forehead Frontalis

Contraction of the frontalis muscle leads to wrinkles on the forehead. This muscle is the only elevator muscle of the upper face originating from the galeal aponeurotic and is inserted into the subcutaneous level and deep dermis of the skin on the superciliary arch. The forehead has high variability due to the diverse individualized animation patterns during facial expressions. The key is to leave the forehead with little activity without leaving a frozen look. Intracutaneous injection of 15–20 U should be targeted at least 1–2 cm above (higher in aged individuals) the orbital rim to prevent brow ptosis (Figure 1). Lateral injection points should extend 2 cm laterally from the eyebrow to minimize excessive lifting of the eyebrow that may lead to a “samurai” or “Spock” look [19,56]. The lateral border of the frontalis muscle is situated 1 cm lateral to the superior temporal crest, which is on the same perpendicular plane as the lateral eyebrow tail [65]. When the BoNT injection point is too medial, diffusion into the lateral portion of the frontalis muscle is difficult and may cause such a look.

#### 3.4.2. Glabella

Wrinkles in this area are formed by continuous action of the procerus and corrugator supercilii muscles, which are brow depressors. They are formed as a result of frowning and may exist as static vertical lines if left untreated in the dynamic state. The corrugator supercilii originates from the medial supraorbital ridge, thickening and thinning laterally [55]. The muscle lies below the frontalis muscle and injections should be intramuscular and deep. The procerus muscle originates from the periosteum of the nasal bone and upper lateral cartilages, and inserts into the glabellar skin and fibers of the frontalis muscle [56].

In total, 12 U to 20 U of BoNT should be injected into 3 to 5 points (Figure 2). Three injection points are usually required in Asian women. Additionally, five-point injections are helpful in men or patients with bulkier glabellas [28]. The injection point for the procerus muscle should be marked at the intersection of the two lines drawn from the medial brow to the contralateral medial canthus. Injection into the corrugator supercilia is deeper in the medial part and slightly superficial on the lateral side. To avoid brow ptosis, injections should not pass through the mid-pupillary line. Injections should be 1 cm superior to the upper inner boundary of the orbital rim to minimize blepharoptosis. This occurs when toxin spreads to the levator palpebrae superioris muscle, making apparent disfigurement and movement of the eyes difficult. The needle should also point upward at a 30° angle [56,66].

#### 3.4.3. Lateral Canthal Lines (Crow’s Feet)

Crow’s feet are caused by contraction of the superficially situated orbicularis oculi muscle. It is a sphincter muscle encircling the orbit and is divided into three portions: palpebral, orbital, and lacrimal [33]. The muscle originates from the nasal part of the frontal bone, the medial palpebral ligament, and the frontal process of the maxilla. The most peripheral preorbital part causes protrusion of the eyebrow and voluntary eye closure is the target of BoNT [67]. Wrinkles in this area can be seen starting in patients aged approximately 20–25 years old [19]. To minimize complications, 6 U to 12 U of BoNT per side should be injected intracutaneously, producing a wheal (Figure 3). Three injections should be targeted 1.5 cm lateral to the lateral canthus or 1 cm outside the bony orbital wall to reduce the chances of diplopia, ectropion, and drooping of the lower eyelid [33]. To prevent asymmetrical smiles, injections should not be targeted close to the inferior margin of the zygoma. This area is susceptible to bruising due to the high vascularity [35,49].

#### 3.4.4. Infraorbital Rhytides

The palpebral part of the lower eyelid closes the eyelid and is subdivided into preseptal and pretarsal portions [49]. Subdermal injection of 2 U to 4 U of BoNT should be administered into the junction of the pretarsal and preseptal portions of the orbicularic oculi (Figure 4). To possess a “charming roll” in the pretarsal area, which makes the patient appear youthful, injections should be distanced from the lower ciliary margin [28]. Medial infraorbital BoNT injection doses should be minimal and injection delicate to prevent lower eyelid edema [28]. Patients with a positive snap test, lower eyelid puffiness, and previous lower lid blepharoplasty are not good candidates because scleral show may result in complications [49].

#### 3.4.5. Bunny Lines

Bunny lines are created by contraction of the transverse nasalis muscle, which draws the nose up and medially [50]. Other muscles such as the procerus, levator labii superioris alaeque nasi (LLSAN), and medial fibers of the orbicularis oculi may also have minor contributions. Intramuscular BoNT injections of 3 U to 4 U should be performed slightly lateral to the mid nose bridge (Figure 5). A third medial injection on the bridge of the nose can be performed in severe cases [50]. Injections that are too lateral or inferior may lead to lip drooping if BoNT is diffused into the levator labii superioris (LLS), levator anguli oris, or LLSAN muscles [67].

#### 3.4.6. Gummy Smile

A symmetrical smile with 1–2 mm gingival exposure is perceived as an esthetic smile [68,69]. Gummy smile is defined as an excessive gingival display greater than 3 mm when smiling. Mazzuco and Hexsel [70] further subclassified the gummy smile into four types: anterior, posterior, mixed, and asymmetric. In moderate gummy smile, the LLSAN elevates and everts the upper lip, while the depressor septi nasi muscle pulls the nasal tip down. In severe gummy smile, the LLS and to a lesser extent the zygomaticus minor (ZMi) also raise the upper lip. A single-point injection of 2 U on each side, 1 cm lateral to the nostril ala, also known as the Yonsei point, can target these three muscles (Figure 6) [56,71].

#### 3.4.7. Perioral Rhytides

Dynamic vertical rhytides in this area are due to sphincter action of the orbicular oris muscle. The orbicularis oris originates from the modiolus, mandible, and the maxilla near the incisor fossa and inserts into the skin of lips encircling the upper and lower lips [67]. The treatment of vertical perioral wrinkles requires patient to contract the lips. Depending on anatomy, intracutaneous and intradermal injections can be administered. In total, 2–3 U of BoNT should be injected along the vermilion border (Figure 7). Flattening of the vermilion border can be a secondary complication due to contraction of the orbicularis oris. However, this can be easily corrected using dermal fillers. Since functional competency, facial asymmetry, lip depression, or lip elevation might occur, low doses of BoNT are recommended [44].

#### 3.4.8. Masseter Hypertrophy

The masseter muscle is one of the four masticatory muscles that facilitates chewing and causes mouth closing. It is a superficial quadrangular muscle that originates from the zygomatic arch and attaches to the lateral border and the angle of the mandible [56]. Particularly in Asian women, a small oval face with a non-square angle and smooth jaw contour is perceived as esthetically appealing. Since hypertrophy of the masseter leads to a broad face, reduction in the muscle using BoNT has become a popular procedure in Asian countries.

Injections need to be both deep and superficial to effectively reduce the large muscle because a deep inferior tendon (DIT) can block diffusion into both levels, causing paradoxical masseteric bulging [65]. Typically, 25 U to 30 U of BoNT can be injected on each side of the masseter (Figure 8). Injection points can be modified in smaller muscles and skinnier patients to minimize diffusion of the toxin anteriorly, which may leave a “sunken cheek” appearance. Injection too anterior or superiorly can also cause “sunken cheek”. Deeply injecting below a line drawn from the mouth corner to the tragus and at least 1 cm away from the anterior border of the masseter muscle is advisable [72,73]. Injection into the masseter can also reduce bruxism, clenching, and myofascial pain symptoms. However excessive doses of BoNT may temporarily cause difficulty in chewing.

#### 3.4.9. Parotid Gland Hypertrophy

The parotid gland is an inverted-pyramidal-shaped organ that lies on the posterior border of the mandible angle [56,74]. Due to congenital, iatrogenic, or acquired conditions, BoNT in the glands plays a role in the esthetically slimming volume of the hypertrophic glands. BoNT can block not only the acetylcholine from binding in the neuromuscular junction but also in the salivary gland [28,75]. Initially, BoNT was used to treat sialorrhea. However, injection to reduce benign hypertrophy of the salivary glands has also been reported [65]. When the mandibular contour is bulky and extends below the angle, parotid gland hypertrophy could be suspected. Since the parotid gland lies superficial to the masseter muscle, 20 U to 30 U of BoNT should be injected more superficially than the masseter (Figure 9) [28]. Since the majority of salivary production comes from the submandibular gland, high-dose injections simultaneously into both the parotid gland and submandibular gland in patients may lead to severe dry mouth [56,76,77].

#### 3.4.10. Drooping Oral Commissures

The depressor anguli oris (DAO) muscle originates from the oblique line of the mandible and inserts into the modiolus [4,66]. Excessive contraction of the depressor anguli oris muscle pulls the mouth corners down creating a drooping and sad impression. Subdermally or intradermally 4 U to 10 U of BoNT can be injected, 1 cm lateral and lower to the mouth corner (Figure 10) [51]. In Asian patients, since the modiolus is situated in a lower position than in Caucasians, injection can be even lower [56]. Injection into the depressor labii inferior could cause difficulty in sipping water, while imprecise targeting of the muscles can lead to an asymmetrical smile.

#### 3.4.11. Cobble Stone Chin

The mentalis muscle is a lower lip levator, which changes the chin texture when hyperactivated. This muscle originates from the alveolar bone inferior to the lateral incisor and attaches medially towards the skin, forming a dome-shaped chin [56]. A pebbled or dimpled appearance of the chin can be treated with two injections, totaling 10 U into the mentalis muscle (Figure 11). Injection too superiorly in proximity to the lower lip can spread BoNT to the orbicularis oris, and too laterally to the DAO. BoNT diffusion into both muscles may lead to ptosis or asymmetry of the lower lip [51]. It is important to remain at least 1 cm below the mental sulcus to avoid oral incompetence [55].

### 3.5. Ultrasonography (US)-Guided Botulinum Toxin Techniques

To maximize accuracy and minimize complications, US-guided treatments have been actively utilized in minimally invasive esthetic treatments, such as BoNT, fillers, and threads [65,78,79]. US is useful for distinguishing between anatomical structures such as vessels, muscle, fat, bone, and glands [66]. Not only can anatomical landmarks be seen in real time, but also injection needles and cannulas can be visualized during the procedures, which make outcomes more predictable (Figure 12). Therapy can be reproducible, which increases the efficacy and safety of esthetic treatments. Rather than injecting blindly into a muscle or layer, BoNT can accurately target a selected area and thereby treat the asymmetry in patients with crooked smiles or even deviated lower faces [80,81].

### 3.6. Complications

BoNT is generally considered to have a wide safety margin in terms of risk. However, since its use has become more common, various injection-related adverse effects (AE) have arisen. Complications are usually technique-dependent, and their incidence decreases as the injector becomes experienced [57]. Complications are the result of reactions to the injection itself or due to the spread of BoNT to untargeted areas [57,82]. Injection-related reactions, such as erythema, edema, and injection pain, are usually mild and temporary and resolve within days. Bruising and ecchymosis are common but tend to take longer to resolve. A recent systematic review reported complication rates in the upper face to be 16% [83]. Headaches and migraines were the most frequently reported AE and this may be related to periosteal trauma or intramuscular hematoma formation during injection [83,84]. Anxiety, vasovagal episodes, paresthesia or dysesthesia, and allergic reactions should also be closely monitored [4,57]. Undesired effects due to the distant spread of BoNT include blepharoptosis, eyebrow ptosis, undesired eyebrow shape, unnatural face, eye opening difficulty, and unsatisfactory results [73,85]. Temporary blepharoptosis is seen in approximately 1–5% of patients [4]. Nasopharyngitis, sinusitis, blurred vision, upset stomach, voice changes, restricted mouth opening, muscle atrophy, and stiffness were also adverse events (AEs) encountered after injection [9,22,84].

The development of neutralizing antibodies is a crucial AE. BoNT may be regarded as a foreign substance in the body that induces an immune response after repeated treatment [86]. Therefore, to reduce antigen occurrence, a commercial product with a low risk of antigenicity should be selected, such as that with fewer complex proteins. Another way to minimize the risk of neutralizing antibody formation is to use the minimum dose needed for effect, injected at least 3 months apart, and avoid booster injections within one month [87,88].

## 4. Discussion

Several expert consensus statements have also been published from various parts of the world including Europe, Pan-Asia, and the Americas [28,33,34,35,38,44,45,46,47,48,49,50,51,52,53]. The results are shown in Table 1, Table 2 and Table 3 and in the Appendix A. There are also numerous systematic reviews related to dosing and effective injections of BoNT. A Cochrane systematic review by Camargo et al. [54] in 2021 looked at 14,919 people (mostly women) from 65 RCTs to see if BoNTA was safe and effective in facial wrinkles. Different types of BoNT were compared with a placebo or each other in cases where glabellar lines, crow’s feet, perioral lines, and forehead lines were treated. The study found that at 4 weeks after injection, all types of BoNT reduced glabellar lines more than the placebo. Regarding safety, ptosis was the only major AE reported and it was seen in fewer than 5% of the trials [54]. The dose range according to the facial region and BoNT brand was also investigated. The dose for glabella lines using onabotulinumtoxinA was 8 U to 80 U, abobotulinumtoxinA was 20 U to 75 U, incobotulinumtoxinA was 20 U to 24 U, daxibotulinumtoxinA was 20 U to 60 U, NewBoNTA (Medytox^®^, Prostigne^®^, Neuronox^®^) was 20 U, LiquidBoNTA was 20 U to 75 U, and prabotulinumtoxinA was 20 U to 60 U [89,90,91]. The dose for forehead lines using onabotulinumtoxinA was 10 U to 48 U, and incobotulinumtoxinA was 10 U to 20 U [92,93]. For crow’s feet the dose using onabotulinumtoxinA was 7.5 U to 24 U, abobotulinumtoxinA was 30 U, incobotulinumtoxinA was 7.5 U to 12 U, and Neuronox^®^ was 24 U [90,94]. Perioral lines were treated with 7.5 U to 12 U of onabotulinumtoxinA [95,96]. The distribution of injection points for the glabellar lines were three to seven points. It was four to eight for the forehead, three points for the crow’s feet, and four points for the perioral lines. The dose and injection areas seem to differ for all types of BoNTAs for each area. Some BoNTA types have similar efficacy, but some areas require more Units for similar effect. Still, the data comparing different types of BoNT are limited, and with the evolution of newer BoNT brands, further investigation is needed to find the accurate efficacy and dose.

The injections points vary slightly between culture and ethnicity, and with sex. In general, men need greater amounts of BoNT injection because they tend to have thicker skin, more facial movement, and larger and stronger muscles [97]. Injection doses for the masseter muscle are higher in the Asian population, and this may be due to slightly different standards of beauty, with Asians preferring a slimmer jaw line [27]. Since most of the AEs occur when higher doses have been injected, it is important to comprehend the adequate technique [54,56]. One interesting finding is that parotid gland hypertrophy injections for cosmetic reasons have only recently been reported in a consensus for Asians [28]. The treatment itself may be preferred in Asian countries where a bulky jaw is perceived as less attractive, and this can be caused by either a hypertrophic masseter muscle or hypertrophic parotid gland. The indications for BoNT are broadening in both esthetical and therapeutical fields. This may be due to adjunctive tools, such as US, that help in precise injection and monitoring of subsequent changes [65]. To date, BoNT has shown promising treatment modalities in many fields, and further research will render more accurate injection techniques and versatile applications.

## 5. Concluding Remarks

There has been a global rise in the demand for non-invasive procedures to help patients rejuvenate and appear youthful. BoNT is now the most performed esthetic procedure in the world with various types of BoNT that have different properties out in the market. More practitioners are using BoNT and demand appropriate guidelines so that injections using BoNT can be performed safely and effectively for the treatment of dynamic rhytides and hypertrophic structures. This paper is a narrative review of the consensus statements of expert practitioners and the various literature on the injection points and techniques, highlighting both the Asian and Caucasian population separately. This paper can serve as a practical illustrative guide and reference for optimal, safe injection areas and effective doses for application of BoNT in the face and oral and maxillofacial area. One limitation of this study was that a single author extracted the data and a systematic meta-analysis was not performed, and therefore the risk of bias was not investigated. Larger consensus statements and dose-ranging systematic studies comparing female to male, Caucasian to Asian, and younger to older patients should be conducted in the future.

## Figures and Tables

**Figure 1 toxins-15-00082-f001:**
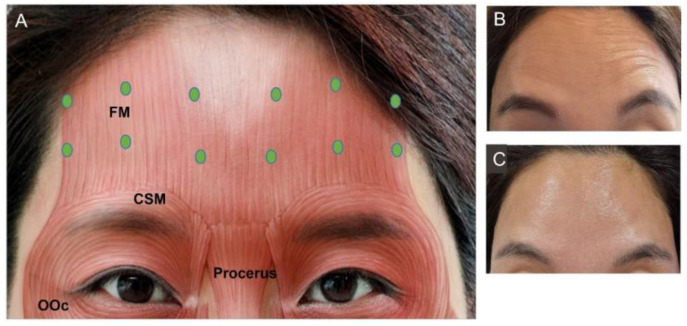
Frontalis muscles. (**A**) Injection points (green dot), facial photographs (**B**) before injection, (**C**) after injection. FM, frontalis muscle; OOc, orbicularis oculi; CSM, corrugator supercilia.

**Figure 2 toxins-15-00082-f002:**
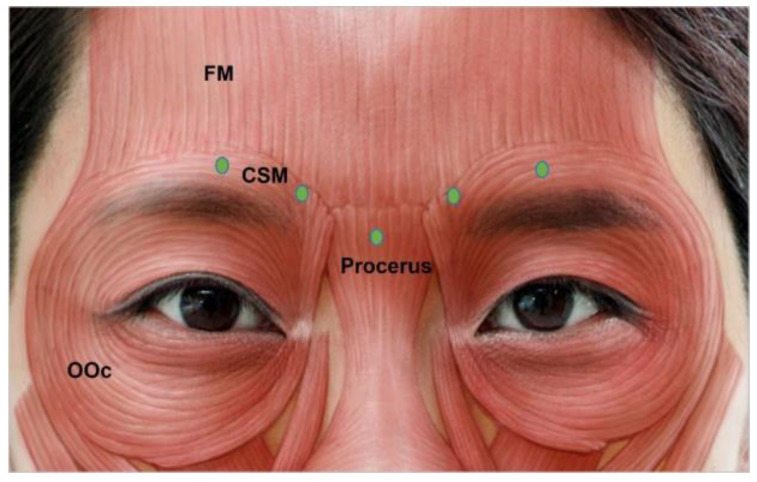
Glabella injection points. FM, frontalis muscle; OOc, orbicularis oculi; CSM, corrugator supercilia.

**Figure 3 toxins-15-00082-f003:**
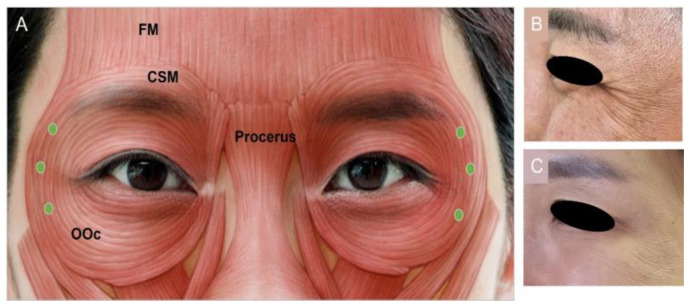
Lateral canthal (**A**) injection points, (**B**) before injection, (**C**) after injection. FM, frontalis muscle; OOc, orbicularis oculi; CSM, corrugator supercilia.

**Figure 4 toxins-15-00082-f004:**
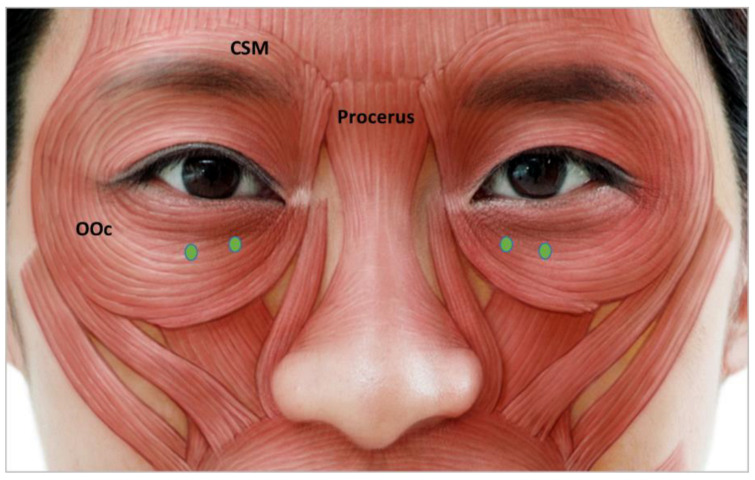
Infraorbital rhytides injection points. OOc, orbicularis oculi; CSM, corrugator supercilia.

**Figure 5 toxins-15-00082-f005:**
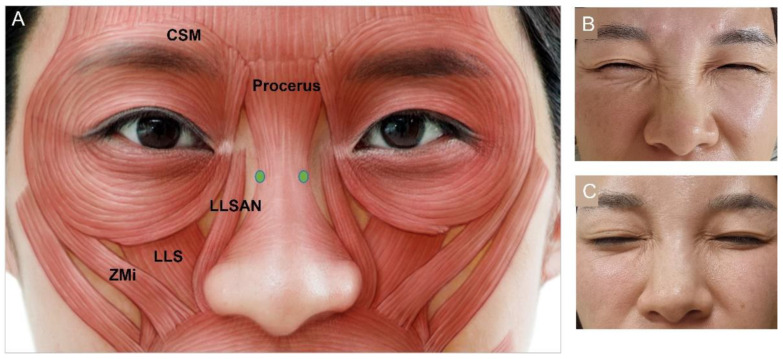
Bunny lines (**A**) injection points, (**B**) before injection, (**C**) after injection. LLSAN, levator labii superioris alaeque nasi; LLS, levator labii sup; Zmi, zygomaticus minor; CSM, corrugator supercilia.

**Figure 6 toxins-15-00082-f006:**
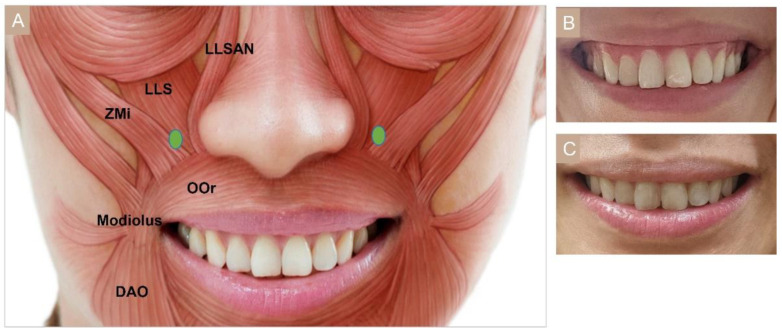
Gummy smile (**A**) injection points, (**B**) before injection, (**C**) after injection. LLSAN, Levator labii superioris alaeque nasi; LLS, Levator labii superioris; Zmi: zygomaticus minor; OOr: orbicularis oris, DAO: depressor anguli oris.

**Figure 7 toxins-15-00082-f007:**
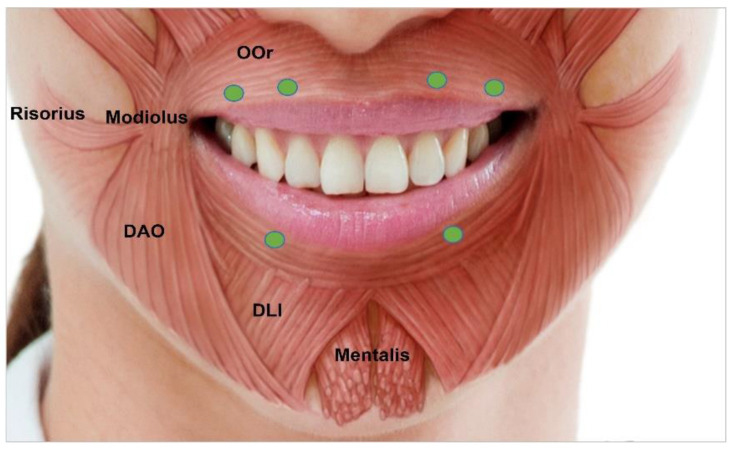
Perioral rhytides injection points. OOR, orbicularis oris; DAO, depressor anguli oris; DLI, depressor labii inferioris.

**Figure 8 toxins-15-00082-f008:**
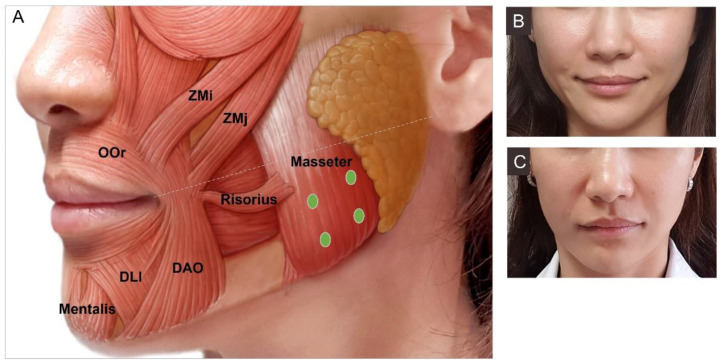
Masseter (**A**) injection points, (**B**) before injection, (**C**) after injection. Zmi, zygomaticus minor; ZMj, zygomaticus major; OOr, orbicularis oris; DAO, depressor anguli oris; DLI, depressor labii inferioris.

**Figure 9 toxins-15-00082-f009:**
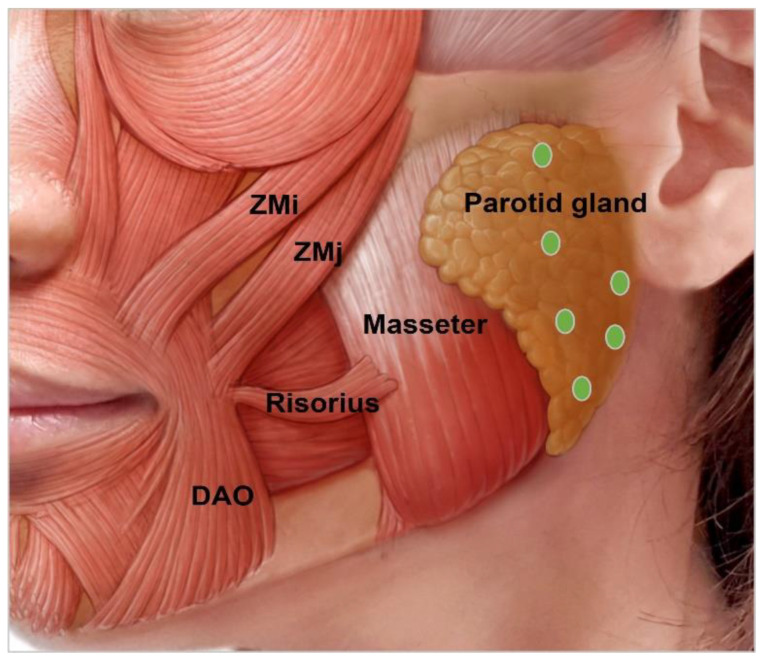
Parotid gland injection points. Zmi, zygomaticus minor; ZMj, zygomaticus major; DAO, depressor anguli oris.

**Figure 10 toxins-15-00082-f010:**
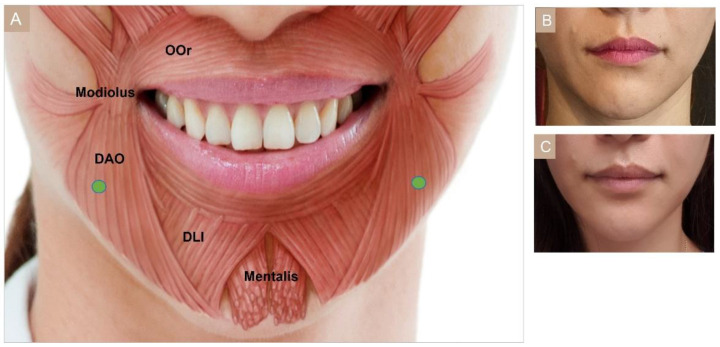
Depressor anguli oris (DAO) (**A**) injection points, (**B**) before injection, (**C**) after injection. OOr, orbicularis oris; DAO, depressor anguli oris; DLI, depressor labii inferioris.

**Figure 11 toxins-15-00082-f011:**
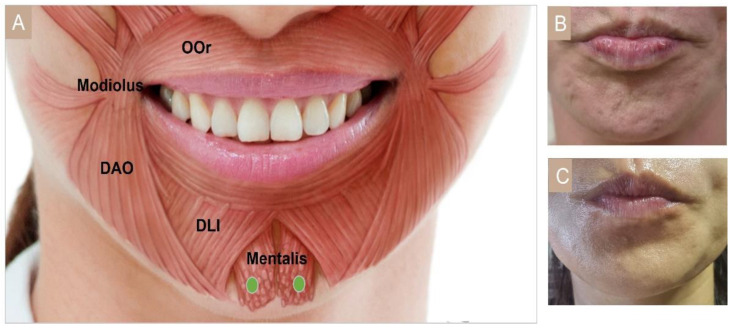
Mentalis (**A**) injection points, (**B**) before injection, (**C**) after injection. OOr, orbicularis oris; DAO, depressor anguli oris; DLI, depressor labii inferioris.

**Figure 12 toxins-15-00082-f012:**
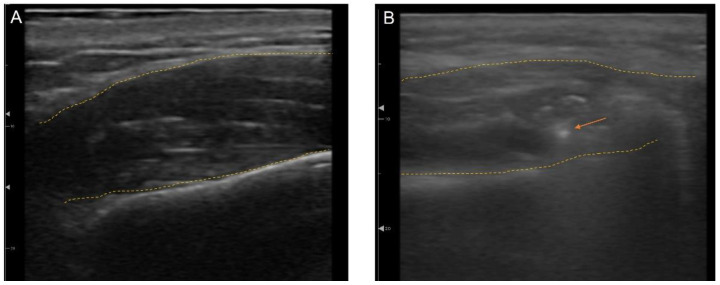
Ultrasonography imaging with outline of the muscle (yellow dotted line) (**A**) masseter, (**B**) masseter with needle tip (orange arrow).

**Table 1 toxins-15-00082-t001:** Consensus recommendations using Botulinum toxin A for the cosmetic treatment of various indications in the upper face.

Authors	Indication									
	Forehead		Glabella		Crow’s Feet		Infraorbital		Bunny Lines	
	Total Dose (U)	IP (n)	Total Dose (U)	IP (n)	Total Dose (U)	IP (n)	Total Dose (U)	IP (n)	Total Dose (U)	IP (n)
	Dose per IP (U)		Dose per IP (U)		Dose per IP (U)		Dose per IP (U)		Dose per IP (U)	
Carruthers et al., 2004, 2008 [33,34]	F: 6–15 M: 6–15	4–8	F: 10–30 M: 20–40	5–7	F: 10–30 M: 20–30	2–5/side	NR	NR	2–5	1/side
Carruthers et al., 2013 [35]	5–15	4–10	F: 10–50 M: 20–60	5–7	10–30	2–5/side	NR	NR	NR	NR
Raspaldo et al., 2011 [38,39]	F: 10–20 M: 10–30	3–5	8–25	2–5	6–16/side	3–4/side	2–4	1–2/side	4–8	1/side
	1–2		4–5		2–4		0.5–2		2–4	
Maas et al., 2012 [44]	11.75 (range: 5–30)	6	20 (range: 12–25)	5	20 (range: 6–30)	3/side	4 (range: 1.5–8)	1/side	6 (range: 2–12)	1/side
Imhof et al., 2013 [45]	F: 8 M: 12	4–6	20	5	4–6	3/side	2	2/side	2	1/side
	0.5–2		4		1–3		0.5		1	
Lorenc et al., 2013 [46]	F: 10–20 M: 20–40	4–6	20 (range: 10–40)	5	8–16/side	4/side	1 (range: 0.5–2.5)	1/side	NR	NR
Yuskovskaya et al., 2015 [43]	10–20	4–8	20	6	12/side	3/side	NR	NR	4–8	1/side
	2–4		2–5		4				2–4	
Sundaram et al., 2016 [47,48]	8–25	4–8	12–40	3–7	6–15/side	1–5/side	0.5–2/side	1–3/side	4–8,10	2–3
	2–4		2–4		1–4		0.5–2		2–4	
De Maio et al., 2017 [49,50,51]	NR	5–7	NR	5	NR	3/side	NR	1/side	4–6	2 or 3
									2	
Kaminer et al., 2020 [52]	F: 2–20 M: 4–30	2–12	F: 6–35 M: 8–40	3–8	6–20	2–6/side	NR	NR	2–10	1–2/side
Signorini et al., 2022 [53]	20	5–10	30	5–7	18	3/side	1–2	1/side	4–6	1/side
	2–4		6		6		0.5–1		2–3	

U, unit; IP, injection point; n, number; F, female; M, male; NR, not reported; Up, upper lip; L, lower lip.

**Table 2 toxins-15-00082-t002:** Consensus recommendations for patients using BoNT for the cosmetic treatment of various indications in the lower face.

Authors	Indication											
	Perioral		Gummy Smile		Masseter Hypertrophy		Parotid Gland Hypertrophy		DAO		Cobble Stone Chin	
	Total Dose (U)	IP (n)	Total Dose (U)	IP (n)	Total Dose (U)	IP (n)	Total Dose (U)	IP (n)	Total Dose (U)	IP (n)	Total Dose (U)	IP (n)
	Dose per IP (U)		Dose per IP (U)		Dose per IP (U)		Dose per IP (U)		Dose per IP (U)		Dose per IP (U)	
Carruthers et al., 2004, 2008 [33,34]	4–5	2–6	2–4	1/side	25–30/side	NR	NR	NR	2–5	1/side	4–5	1–2
Carruthers et al., 2013 [35]	4–6	2–6	NR	NR	NR	NR	NR	NR	1–15	1/side	4–10	1–2
									1–7.5			
Raspaldo et al., 2011 [38,39]	Up: 2–5 L: 2–5	Up: 2–4 L: 2–4	3–5/side	1/side	18–30/side	1–5/side	NR	NR	2–5/side	1–2/side	6–10	2
	~1		1–2		6				1–2.5		3–5	
Maas et al., 2012 [44]	5 (range: 2–12)	Up: 4 L: 2	5 (range: 2–10)	1/side	32.5 (range: 20–60)	3/side	NR	NR	6 (range: 2–10)	1/side	5 (range: 3–10)	2–3
Imhof et al., 2013 [45]	4	Up: 4 L: 2	NR	NR	4–6	2 or 3	NR	NR	1–3	1/side	6 (range: 2–8)	2
					2							
Lorenc et al., 2013 [46]	5	2	NR	NR	NR	NR	NR	NR	5	1/side	4–5	2
	2.5								2.5			
Yuskovskaya et al., 2015 [43]	Up: 4–6	Up: 4	NR	NR	50	3/side	NR	NR	6	1/side	6	2
	1–1.5				2				3		3	
Sundaram et al., 2016 [47,48]	1–5	2–5	1–4, 8	1–2/side	15–40	1–5/side	NR	NR	2–4/side	1–2/side	4–10	1–4
	0.5–1		0.5–2		5–15				2		2–3	
De Maio et al., 2017 [49,50,51]	Up: 2–4 L: 2	U: 2–4 L: 2	6–10	3–5	6–24/side	3/side	NR	NR	4–8	1/side	4–8	1- 3
	1		2		4–8				2–4		2.5	
Kaminer et al., 2020 [52]	2–10	2–8	NR	NR	5–35	2–6/side	NR	NR	2–6	1–2/side	2–10	1–4
Signorini et al., 2022 [53]	1–4	Up: 2–4	6–10	3	25–50/side	3–5/side	NR	NR	4–8	1/side	8–10	1–2
	0.5–1		2		5–10 or 7–22				2–4		4–10	

U, unit; IP, injection point; n, number; F, female; M, male; NR, not reported; Up, upper lip; L, lower lip.

**Table 3 toxins-15-00082-t003:** Consensus recommendations for Asian patients using BoNT for the cosmetic treatment of various indications.

Indication	Authors											
	Ahn et al., 2013 [27]			Wu et al., 2016 [30]			Sundaram et al., 2016 [28]			Kapoor et al., 2017 [29]		
	Total Dose (U)	Dose per IP (U)	IP (n)	Total Dose (U)	Dose per IP (U)	IP (n)	Total Dose (U)	Dose per IP (U)	IP (n)	Total Dose (U)	Dose per IP (U)	IP (n)
Forehead rhytides	6–13.5	1–1.5	6–9	5–12	1–4	NR	12–14 (range: 2–32)	0.1–5	12	F: 6–8 M: 10–12	NR	4–6
Glabella	8	4	3	12–20	NR	3–10	10–20 (range: 6–25)	2–4	3–5	16–20	NR	5–7
Crow’s feet	14	2–3	3/side	6–12/side	1–4	NR	6–9 (range: 4–16)/side	2–4	3–4/side	8–12/side	NR	3–4/side
Infraorbital rhytides	1–2	0.5	2–4	4–6	NR	NR	1–2	0.5–1	2–3/side	n/a	NR	NR
Bunny lines	6	2	3	4–6	NR	NR	3–4	0.5–5	2	2–4/side	1–2	2
Perioral rhytides	0.8–1.2	0.2–0.3	4	2–8	NR	NR	2–3 (range: 1–8)	0.5–1	2–6	2–4	NR	Up: 4 L: 2
Gummy smile	4–6	2–3	1/side	2–12	NR	NR	2–4	1–2	2	2–3/side	NR	2
Bulky jaw	24–30/side	8–10	3/side	40–80	NR	NR	20–40	4–6	3–5/side	15–30/side	NR	3/side
Parotid gland hypertrophy	NR	NR	NR	NR	NR	NR	20–40/side	4–6	4–6	NR	NR	NR
DAO	6–10	3–5	1/side	4–6	NR	NR	2–4/side	2–4	1/side	2–3/side	NR	1/side
Cobblestone chin	mild: 5 severe: 10	5	mild: 1 severe: 2	4–8	NR	NR	8 (range: 2–16)	2–4	1–2/side	6–8	NR	1–2

U, unit; IP, injection points; n, number; F, female; M, male; NR, not reported; Up, upper lip; L, lower lip.

**Table 4 toxins-15-00082-t004:** Summary of cosmetically used BoNT. The active substances, excipients, and stabilization form per vial, and the commercial brand names are shown.

US FDA-Approved Generic Name	Active Substance	Excipient	Unit/ Vial	Stabilization (form)	Commercial Products
OnabotulinumtoxinA	Complex (900 kDa)	HSA 0.5 mg, NaCl 0.9 mg	100	Vacuum-dried (P)	Botox^®^, Botox Cosmetic^®^, Vistabel^®^, Vistabex^®^
AbobotulinumtoxinA	Complex (500–900 kDa)	HSA 0.125 mg, Lactose 2.5 mg	500	Lyophilized (P)	Dysport^®^, Reloxin^®^, Azzalure^®^
IncobotulinumtoxinA	Complex-free (150 kDa)	HSA 1 mg, Sucrose 4.7 mg L-methionine (q.s.)	100	Lyophilized (P)	Xeomin^®^ Bocouture^®^
LanbotulinumtoxinA	Complex (900 kDa)	Gelatin 5 mg, Dextran 25 mg, Sucrose 25 mg	100	Lyophilized (P)	BTXA^®^, Prosigne^®^, Lantox^®^
LetibotulinumtoxinA	Complex	HSA 0.5 mgNaCl 0.9 mg	100	Lyophilized (P)	Botulax^®^, Regenox^®^, Zentox^®^
PrabotulinumtoxinA	Complex (900 kDa)	HSA 0.5 mgNaCl 0.9 mg	100	Vacuum-dried (P)	Nabota^®^, Jeuveau^®^, Nuceiva^®^
N/A	Complex (940 kDa)	HSA 0.5 mgNaCl 0.9 mg	100	Lyophilized (P)	Meditoxin^®^, Neuronox^®^
N/A	Complex (900 kDa)	Gelatin 6 mg, Maltose 12 mg	100	Lyophilized (P)	Relatox^®^
N/A	Complex	Polysorbate	25	L	Innotox^®^
N/A	Complex-free (150 KDa)	Polysorbate 20 (q.s.) Sucrose 3 mg NaCl 0.9 mg	100	Lyophilized (P)	Coretox^®^
DaxibotulinumtoxinA	Purified toxin (150 kDa)	Peptide (RTP004) Polysorbate 20 sugar	100	Lyophilized (P)	DAXXIFY^®^
RimabotulinumtoxinB	Complex (700 kDa)	HSA 0.5 mg NaCl 5.844 mg Sodium succinate 1.621 mg	5000	L	Neurobloc^®^, Myobloc^®^

HSA, Human Serum Albumin; NACL, sodium chloride; P, powder; L, liquid; q.s., quantum sufficit.

**Table 5 toxins-15-00082-t005:** Facial treatment areas in correlation with the target muscles, total amount of botulinum toxin, and number of injection points.

Facial Treatment Areas	Target Muscles	Total Dose	Dose per IP	Number of IP
Forehead rhytides	Frontalis	6–20 U	0.5–1 U	4–6/row (2 rows)
Glabella (Frown lines)	Corrugator supercilia, Procerus, Depressor supercilii	12–30 U	2–4 U	3–5
Crow’s feet (Lateral canthal lines)	Lateral orbicularis oculi	4–16 U/side	2–4 U	3–4/side
Infraorbital rhytides	Orbicularis oculi	2–4 U	0.5–1 U	2–3/side
Bunny lines (Nasal rhytides)	Nasalis (transverse) Levator labii superioris alaeque nasi	3–4 U	1.5–2 U	2
Perioral rhytides (lipstick lines, smoker’s lines)	Orbicularis oris	2–3 U	0.5 U	4–6
Gummy smile	Levator labii superioris, Levator labii superioris alaeque nasi, Zygomaticus minor	2–6 U	1–2 U	1–2/side
Bulky jaw	Masseter	15–30 U/side	5 U	3–5/side
Hypertrophic parotid gland	Parotid gland	20–40 U	4–6 U	4–6/side
Drooping oral commissures (frown lines, DAO)	Depressor anguli oris	4–10 U	2–5 U	2
Cobblestone chin	Mentalis	8–10 U	2–4 U	2

U, unit; IP, insertion point. Numbers in the table are adapted from consensus literature summarized in Table 1, Table 2 and Table 3.

## Data Availability

Not applicable.

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
