# Peer review of "Cosmetic Treatment Using Botulinum Toxin in the Oral and Maxillofacial Area: A Narrative Review of Esthetic Techniques"

_toxins, 2023, doi:10.3390/toxins15020082_

Round 1

Reviewer 1 Report

Thank you for the opportunity to review your manuscript.  I have no major issues with the content of the paper.  There are a few grammatical changes that I would recommend:

Line 109-11: Unclear what the Mephisto look is or how injection lateral to the brow prevents it.

Line 159: m abbreviation should be spelled out unless always abbreviated.

Line 184-6: I don't understand this sentences

Line 192: ...causes "the" mouth to close

Lines 274-278:  duplicated sentence

Line 220:"...can be considered" rather than "debated", though I am not certain that is true.  Reference?

Lines 225-6: marionette lines are not corrected with botox as they are not due to muscle contraction.  Rather, injection into the DAO provides a lifting of the commissure due to unopposed upward pull of the commissure elevators.

Author Response

Dear reviewer thank you for your keen observation and recommendations.

I only had a week to revies, but I tried my best to revise the paper as much as I could to meet your requests. Thank you for the precious time and effort you have given to look over this paper.

Below are the remarks you gave and my answers.

  1. Line 109-11: Unclear what the Mephisto look is or how injection lateral to the brow prevents it.

Author's Answer: The “Mephisto” is a term that has not been used often recently, so I revised it to a “samurai” or “Spock” look. Lateral injection points should extend 2cm laterally from the eyebrow to minimize excessive lifting of the eyebrow that may lead to a “samurai” look which is caused when BoNT isn’t diffused into the lateral portion of the frontalis muscle. The lateral border of the frontalis muscle is situated 1cm lateral to the superior temporal crest which is on the same pependicular plane as the lateral eyebrow tail. When the BoNT injection point is too medial, diffusion into the lateral portion of the frontalis muscle is difficult and may cause such a look. Jaspers et al stated that injection should be 2cm lateral from the eyebrow to prevent this.

  1. Line 159: m abbreviation should be spelled out unless always abbreviated.

Dear reviewer, thank you for your recommendation. I modified “m” to “muscle” in the text.

  1. Line 184-6: I don't understand this sentences

    Dear reviewer sorry for the grammatical error. Also this section was revised to give the reader a better understanding about the gummy smile. Now it says:

    A symmetrical smile with 1-2mm gingival exposure is perceived as an esthetic smile. Gummy smile is defined as an excessive gingival display greater than 3mm when smiling. Mazzuco and Hexsel further subclassified the gummy smile into four types: anterior, posterior, mixed, and asymmetric. In moderate gummy smile, the LLSAN elevates and everts the upper lip while the depressor septi nasi muscle pulls the nasal tip down. In severe gummy smile, the LLS and to a lesser extent the zygomaticus minor (ZMi) also raise the upper lip. A single-point injection of 2 U on each side, 1 cm lateral to the nostril ala, also known as the Yonsei point, can target these three muscles (Figure.6).

  2. Line 192: …causes "the" mouth to close

Thank you dear reviewer for you suggestions. I sent my paper for a professional English grammar check. I am sorry for such grammatical errors. I modified “that’ to “the”.

  1. Lines 274-278:  duplicated sentence

Dear reviewer, I am so sorry about the duplicated sentence. This sentence was deleted and this section was revised.

  1. Line 220:"…can be considered" rather than "debated", though I am not certain that is true.  Reference?

Dear reviewer, you are totally right. I am so sorry for this. I misunderstood the literature, and rewrote this section. Now it says “Since the majority of salivary production comes from the submandibular gland, high-dose injections simultaneously into both the parotid gland and submandibular gland in patients may lead to severe dry mouth.” The references for this sentence were added as well. Thank you again for checking this section.

  1. Lines 225-6: marionette lines are not corrected with botox as they are not due to muscle contraction.  Rather, injection into the DAO provides a lifting of the commissure due to unopposed upward pull of the commissure elevators.

Thank you reviewer. In some papers they used the term marionette line so I also used this term. But as you point out, the term I used was wrong. It is the DAO that I am treating, and I excluded marionette line from the text and changed it to DAO. Thank you!

Your recommendations have helped this paper phenomenally. Thank you again for your precious time.

Reviewer 2 Report

Dear authors,

I evaluated the article "Cosmetic treatment using botulinum toxin in the oral and maxillofacial area: A review of esthetic techniques". It is a simple review with information about the theme presented.

It was no observed parameters to find the information shown, and I can consider the presence of bias in the content.

Technically, the abstract is poorly presented; the introduction is not showing the "state of the art" of the materials used; Indications and contraindications is incomplete.

The other parts are well-written.

Therefore, where is the discussion and conclusion?

Author Response

Dear reviewer thank you for your keen observation and recommendations.

I only had a week to review, but I tried my best to revise the paper as much as I could to meet your requests. Thank you for the precious time and effort you have given to look over this paper.

Below are the remarks you gave and my answers.

  1. It was no observed parameters to find the information shown, and I can consider the presence of bias in the content.

You mentioned that there were no parameters for the information shown. I am so sorry that I did not include this in my initial paper. I have included the references that I initially organized but did not show on the paper. I included in the table that the parameters were adapted from the expert consensus recommendations from Carruthers et al, Maas et al, de Maio et al, Sundaram et al, and Raspaldo et al. I also included a table comparing the parameters for different treated areas of the face.

  1. Technically, the abstract is poorly presented; the introduction is not showing the "state of the art" of the materials used; Indications and contraindications is incomplete.

Thank you for your helpful remarks. You are absolutely right. The abstract was rewritten to give a better explanation about this simple review. The introduction was also modified as you requested, but I’m not sure if it meets up to your standards. I hope this modifed introduction makes the paper a little more interesting. Additional indications and contraindications for the cosmetic use of botulinum toxin were also rewritten in the paper. Other aspects of the paper were also revised. (Table 2, all sections of the paper)

  1. The other parts are well-written.

Thank you for your compliment.

  1. Therefore, where is the discussion and conclusion?

Thank you for pointing this out. A discussion was added including a review of injection amounts and points of several indication for the face. Also several expert consensus recommendations for injection dose and points were organized into a table (Table.3) in the discussion section.A conclusion was also added at the end of the paper. 

Your recommendations have helped this paper phenomenally. Thank you again for your precious time.

Round 2

Reviewer 2 Report

Dear authors,

I evaluated the article "Cosmetic treatment using botulinum toxin in the oral and maxillofacial area: A review of esthetic techniques". It is a simple review with information about the theme presented, and I appreciate how it was written.

- There was an improvement in this simple review. Otherwise, I considered yet bias in: how to find the information, selection of articles; moreover, the way that it is presented is similar a book chapter and not an article.

- Also, there was not parameters to replicate this study, such as how to find the articles used and how was the pattern used to obtain the data used.

- Technically, the abstract improved but the introduction continues does not showing the "state of the art" of the materials used.

- I am still waiting for the discussion and conclusion session.

Author Response

Dear reviewer,

Thank you with my deepest gratitide for reviewing this paper.

As you pointed out, the reviews I chose in this narrative review were not specifically addressed. To be honest, I initially did not know what you meant by the "state of the art" materials, indications, and contraindications.

I have now included the way my researc can be replicated with a revised discussion and conclusion . I would be so grateful if you could give me some of your precious time to look over the alterations.

In the introduction I have addressed the "state of the art" review method . I have also added more consensus reviews and organized them into 3 tables: Asian consensus, general population consensus on the upper face, general consensus on the lower face.

My paper is not a systematic review, but a narrative literature review of the consensus of botulinum toxin injection methods. I selected the consensus papers using a systematic approach. I have also written my manuscript as so. If you find the word "narrative review with systematic search" confusing, I can adjust this. 

------My introduction added the following--------

This narrative review depicts the anatomy of the facial muscles relative to the BoNT injection points. It is a practical guide tofocuses on esthetic BoNT treatment procedures that both dentists and doctors can perform in the oral and maxillofacial area. The purpose of this A review is to present an overview of BoNT and demonstrate the optimal, safe injection areas and effective dose for application of BoNT of in the face.history, indications, injection techniques, targeted muscles, and some commercially available BoNTs properties will be introduced.

This practical guideline includes a narrative literature review. The review explores quantitative data using a systematic search on studies depicting dose range and injection points from expert consensus opinions. A literature search was conducted using MEDLINE and PubMed electronic databases for the period from January 2002 to December 2022. Manual study searches were also performed using the reference list of articles included during the search process.

The inclusion criteria were as follows:

  1. Studies in the English language
  2. Include dosage of BoNT and/or number of injection points
  3. Be performed in human adults with no diseases
  4. Given BoNT injection of the face for cosmetic or esthetic purpose
  5. Literature limited to expert consensus

Exclusion criteria were as follows:

  1. Studies using BoNT for other reasons than cosmetic/aesthetic: i.e. stroke, neurogenic, bladder management, spasticity, sialorrhea, bruxism, strabismus, myofascial pain, TMJ, obesity
  2. Studies using BoNT for cosmetic/aesthetic reasons in other parts of the body or other reasons: i.e. skin texture, scar, androgenic alopecia, neck wrinkles, trapezius hypertrophy, calf hypertrophy
  3. Studies that investigated only a single muscle

The search included the following keywords used in different combinations: “botox”, “botulinum toxin”, “consensus”, “aesthetic”, “,“cosmetic”, “systematic review”, and “face.” The term “botox consensus” retrieved 202 studies, “botulinum toxin” AND “consensus” retrieved 315 studies. The search term “face botulinum toxin consensus” resulted in 56 studies, “esthetic botulinum toxin” AND “consensus” resulted in 54 articles, and “cosmetic botulinum toxin consensus” resulted in 42 studies. The term “botox systematic review” retrieved 421 articles. Titles and abstracts were screened, the eligibility criteria was applied, and full text analysis were performed in relevant publications. 33 full text articles were screened for eligibility, of which 6 were excluded because of a lack of BoNT dosages and type of paper. [22-24]A total of 27 articles were included in the literature review. The injection dosage and points proposed in this paper are based on the finding from such various reviews. -

---The discussion was also edited------

A Cochrane systematic review by Camargo et al.[27] in 2021 looked at 14,919 people (mostly women) from 65 RCTs to see if BoNTA was safe and effective in facial wrinkles. Different types of BoNT were compared with placebo or each other in cases where glabellar lines, crow’s feet, perioral lines, and forehead lines were treated. The study found that at 4 weeks after injection all types of BoNT reduced glabellar lines more than the placebo. Regarding safety, ptosis was the only major AE reported and it was seen in less than 5% of the trials[27]. The dose ranged according to the facial region and BoNT brand was also investigated. The dose for glabella lines using OnabotulinumtoxinA was 8 U to 80 U, AbobotulinumtoxinA was 20 U to 75 U, IncobotulinumtoxinA was 20 U to 24 U, DaxibotulinumtoxinA was 20 U to 60 U, NewBoNTA (Medytox®, Prostigne®, Neuronox®) was 20 U, LiquidBoNTA was 20 U to 75 U, and PrabotulinumtoxinA was 20 U to 60 U[69-71]. The dose for forehead lines using OnabotulinumtoxinA was 10 U to 48 U, and IncobotulinumtoxinA was 10 U to 20 U[72, 73]. For crow’s feet the dose using OnabotulinumtoxinA was 7.5 U to 24 U, AbobotulinumtoxinA was 30 U, IncobotulinumtoxinA was 7.5 U to 12 U, and Neuronox® was 24 U[70, 74]. Perioral lines were treated with 7.5 U to 12 U of OnabotulinumtoxinA[75, 76]. The distribution of injection points for the glabellar lines were 3 to 7 points. It was 4 to 8 for the forehead, three points for the crow’s feet, and four points for the perioral lines.

Several expert consensus statements have also been published from various parts of the world including Europe, Pan-Asia, and the Americas.[37, 38, 42, 45, 46, 57, 77-85] BoNT dose and injection points of expert opinions are organized summarized in Tables 3 and 4. A total of 27 consensus articles were reviewed after retrieving the data through a systematic search. Of the 27 studies, 4 studies were specified towards Asians, and therefore organized into a separate table.[37, 86-88] Of the 23 studies, a study by Fagien et al[89] and by Bertossi [90] had been each separately updated by the same group and therefore was excluded from the table. Carruthers et al.[42] had updated their consensus in 2008 and later again in 2013. The initial 2004 consensus was grouped with the 2008 consensus, but the 2013 consensus was summarized because of slightly different parameters. A study by Raspaldo et al[91] and Gassia et al[92] was identical to the 2010 consensus from Raspaldo et al.[44, 79] and excluded from the summary tables.  Most of the studies used onabotulinumtoxinA Units but 3 studies using different Units of abobotulinumtoxinA were excluded from the review[93-95]. Nits of one study by Sundaram et al[37] and Yutskovskay et al[96] used incobotulinumtoxinA while Ahn et al[86] utilized Medytox. A total of 16 consensus papers were summarized in this review.  

The iInjections points vary slightly between culture and ethnicity, but in general, men need more amounts of BoNT injection because they tend to have thicker skin, more facial movement, and larger and stronger muscles.[97]. Injection dose for the masseter muscle are higher in the Asian population, and this may be due to slightly different standards of beauty preferring a slimmer jaw line in Asians.[86] Since most of the AEs occur when higher doses have been injected, it’s important to comprehend the adequate technique[27, 52]. One interesting finding is that parotid gland hypertrophy injections for cosmetic reasons has only recently been reported in a consensus for Asians[37]. The treatment itself may be preferred in Asian countries where a bulky jaw is perceived as less attractive, and this can be cause by either a hypetrophic masseter muscle or hypertrophic parotid gland. The indications for BoNT are broadening in both esthetical and therapeutical fields. This may be due to adjunctive tools , such as US, that help in precise injection and monitoring changes.[40] Up to date, BoNT shows promising treatment modalities in many filedsfields and further research will render more accurate injection techniques and versatile applications.

One limitation of this study was that a single author extracted the data and a systematic meta-analysis was not performed, and therefore the risk of bias was not investigated. Larger consensus statements and dose-ranging systematic studies comparing female to male, Caucasian to Asian, and younger to older patients should be conducted in the future.

--The conclusion was also revised----------

There has been a global rise in demand for noninvasive procedures to help patients rejuvenate and appear youthful. BoNT is now the most performed esthetic procedure in the world with various types of BoNT that have different properties out in the market. More practitioners are using BoNT and are in need of appropriate guidelines so iInjections using BoNT can be done safely and effectively for the treatment of dynamic rhytides and hypertrophic structures. This paper is a simple narrative review of the consensus statements of expert practitioners and various literature on the injection points and techniques. The primary aim of this paper was to provide a practical guide and provide optimal, safe injection areas and effective dose for application of BoNT in the face. This paper can aid as a reference for novice physicians and dentists to get an understanding of the indications, anatomical references, injection doses, and injection patterns related to BoNT.

Thank you very much for your time and considerations